# Numerical Simulation of a Lab-on-Chip for Dielectrophoretic Separation of Circulating Tumor Cells

**DOI:** 10.3390/mi14091769

**Published:** 2023-09-15

**Authors:** Abdallah M. Alkhaiyat, Mohamed Badran

**Affiliations:** Department of Mechanical Engineering, The American University in Cairo, New Cairo 11835, Egypt; abdallah.m@aucegypt.edu

**Keywords:** circulating tumor cells, dielectrophoresis, cell separation, lab-on-chip

## Abstract

Circulating tumor cells (CTCs) are cancer cells detached from tumors that enter the bloodstream with the rest of the blood cells before settling on remote organs and growing. CTCs play a major role as a target for cancer diagnosis. This study aims to propose and simulate a lab-on-chip (LOC) design that separates CTCs from white blood cells (WBCs) and blood platelets (PLTs) using low-voltage dielectrophoretic separation with high efficiency. The proposed design include two stages a passive and an active one cascaded in a compact package. Numerical simulations are performed on the COMSOL Multiphysics^®^ software package to optimize the geometric parameters of the LOC, such as the width and length of the microchannel and the number of electrodes and their arrangements. Moreover, the effects of adjusting the applied voltage values as well as buffer inlet velocity are investigated. The proposed LOC design uses four electrodes at ±2 V to achieve 100% separation efficiency for the three cell types in simulation. The 919 µm × 440 µm LOC has a channel width of 40 µm. The inlet velocities for the blood-carrying cells and buffer are 134 and 850 µm/s, respectively. The proposed LOC can be used for the early detection of CTCs, which can be beneficial in cancer diagnosis and early treatment. In addition, it can be used in cancer prognosis, treatment monitoring and personalizing medicine.

## 1. Introduction

Analyses of biological cells often involve the isolation of cells from a biological fluid. Cells are then sorted in preparation for further analysis. For this reason, particle separation is often implemented as an initial step in diagnosis, screening, or treatment pipelines [1]. Two distinct particle manipulation methods that apply to separation exist based on the usage of external electric fields: active and passive separation [2]. Passive separation devices make use of fluid drag forces, inertial forces, and the structure of the microchannel. Active separation, however, depends on the use of external forces to manipulate particles.

Passive separation lab-on-chips (LOCs) have the advantage of being simple and having high throughput [3]. They can sort different particles using filtration or mechanical trapping [4,5], based on the densities of particles [6], based on cell opacity [7], or based on the mass of the particles using centrifugal forces [8,9,10,11]. Active separation methods add complexity to the LOC, require external power, and generally have more limited throughput. However, it allows real-time control of the target cells. Examples of these methods include magnetic force separation [12,13,14,15], and dielectrophoretic force separation [16,17,18,19,20,21,22,23,24,25].

Microfluidic flows are characterized by low Reynolds numbers, lending themselves to separation methods that depend on streamlined flows. Single flows can be separated using dean flow and inertial flow separation in curved and straight microchannels, respectively [26]. Each cell species will follow a particular trajectory based on the cell diameter, the viscosity of the medium and its flow rate. This leads to eventual separation at the end of the microchannel [27,28,29,30]. In contrast, pinched flow fractionation (PFF) [31,32] is a separation method that uses two flows: a sample and a pinch stream (buffer). When the buffer is introduced to the channel, a narrow sample stream is formed. Cells larger than the sample stream follow the interface of the two streams, while smaller cells are forced to follow the sample stream [26]. Previous studies used the pinch channel width to target specific cell species [31]. Special precautions should be taken when optimizing the microchannel geometry as it affects cell deformation and shear rates [32].

Other previous attempts used the Zweifach–Fung effect separation [8,33,34,35,36], also known as the bifurcation law, which relies on the difference in resistance between a main channel and daughter subchannels to extract relatively large cells. This means that the volume fraction of these cells tends to increase in the daughter subchannels with higher flow rates, as opposed to lower-flow-rate main channels. A critical flow rate is achieved at a certain cell-to-channel size ratio [36]. Finally, some designs combine multiple separation techniques for separation. For instance, this can be performed by combining multiple geometries such as trapezoidal and rectangular cross-sections in a spiral microchannel [27] to make use of dean flow and the Zweifach–Fung effect for cell separation. The cross-section in such designs needs to be carefully optimized, including the outer and inner wall heights, the base width, and the slant angle, in addition to the number of turns of the spirals. Alternatively, multiple stages can be used for focusing, and then separation can be performed in a second stage via bifurcation [37].

Circulating tumor cells (CTCs) are cancerous cell clusters in the bloodstream. They can be used in liquid biopsy and are potentially early-detection cancer biomarkers [24,30,38,39]. CTCs are unique compared to other biomarkers in that they have distinct phenotypic and morphological features that can identify the stage and type of cancer [40]. Isolation of CTCs from the other blood components is therefore beneficial in cancer diagnosis, prognosis, and treatment monitoring. Early detection of CTCs is of paramount importance in the screening stage, as it allows treatment before tumor dissemination and metastases. In addition to CTC separation, the ability to separate white blood cells (WBCs) from the same blood sample enables extensive analysis that is beneficial for cancer patients. For instance, the count of WBCs can be indicative of tumor growth in certain types of cancers [41,42,43,44]. The WBC count can also serve as an indication for autoimmune and inflammatory diseases, bacterial or viral infections, as well as potential allergic reactions to treatment. Further integration of WBC sorting stages can be achieved based on cell opacity [16] in conjunction with cell sizes.

The proposed design attempts to attain a high separation efficiency for three species (CTCs, WBCs, and platelets (PLTs)) from a single blood sample in a compact package. The aim is to maintain low voltage values in order to keep potential cell membrane damage to a minimum [45,46]. It is a hybrid design that makes use of an active and a passive stage in order to combine the benefits of both methods while negating most of the shortcomings. This is a novel approach that uses DEP force during the active separation stage for the isolation of WBCs and a passive stage for CTC separation. The design parameters of this two-way separator are then parametrically simulated in order to arrive at an optimum design. 

Achieving the high separation efficiency requires a microchannel based on a multi-Y-channel configuration. The proposed LOC utilizes active separation in the first stage in the form of dielectrophoretic (DEP) forces. DEP force induces the motion of the polarized cells in the blood sample, which is an effective separation method of bioparticles [47], especially with the much lower concentration of CTCs compared to blood cells, estimated to be at 1–10 cells in a single mL of blood [48]. The extreme rarity of CTCs compared to other blood cells, with 1–100 CTCs in a 7.5 mL whole blood CTC [49,50] necessitates the preprocessing of whole blood samples before separation can be practical. For instance, RBCs can be eliminated via density-gradient centrifugation [51], and only the buffy coat containing peripheral blood mononuclear cells (PBMCs) is isolated. The PBMCs (containing WBCs and PLTs) can then be diluted and injected into the microchannel for separation.

Various microfluidic structures can be used for dielectrophoretic separation [52]. In this study, a simple planar microchannel is selected due to the ease of fabrication and modeling. Planar microstructures can be created relatively easily through photolithography and polydimethylsiloxane (PDMS) molding processes [53], etching [54,55,56], or thick-film screen-printing [21,57,58]. Additionally, planar designs are easier to model as they can be simulated in two dimensions. This fact is amplified when optimizing the design geometric parameters, where performing parametric optimization studies is a necessity. In contrast, microchannels that make use of dean forces or diffuser–nozzle effect using non-rectangular channel geometries [8,30,53,59,60] often require the use of three dimensions in order to be simulated.

Herein, the physical and electrical properties of the targeted cells are first introduced. The modeling and simulation procedure is then discussed in detail, and the theoretical background of cell separation using dielectrophoresis is also presented. The last sections report the results, discussion, and conclusions.

## 2. Materials and Methods

A two-stage Y-channel separation LOC is used for the separation of three species: CTCs, WBCs, and PLTs. The modeling and simulation procedure is introduced, including modeling, boundary conditions, mesh refinement, and model verification is presented. Finally, a summary of the simulated combination is presented.

### 2.1. Target Cell Types

Three cell types are considered due to the elimination of RBCs after centrifugation. These are CTCs, PLTs, and WBCs. Table 1 shows the physical and electrical properties [32,61,62] of targeted cells.

### 2.2. Modeling and Simulation Setup

The simulation software package of choice in this study is COMSOL Multiphysics^®^ version 6.0 due to its flexibility in combining the required Multiphysics for modeling the LOC device. These include modeling the fluid flow, modeling the applied electric current, and finally, particle tracing for the three cell species.

#### 2.2.1. Computational Model

The electric field induces the major force responsible for cell manipulation, the dielectrophoretic force. The electric potential and electric field relation:(1)E=−∇·V.
where E is the electric field vector, and V is the electric potential. The current is given by:(2)J=σE,
where σ is the conductivity, and J is the current density.

Creeping flow physics is used to model the fluid flow confined within the LOC’s microchannels using the Navier–Stokes equations, neglecting the inertia term. This type of flow, also known as Stokes flow, is characterized by having small Reynolds numbers. The main equations are:(3)0=∇·[−pI+K]+F,
and:(4)ρ∇·u=0.

Here, pI is the pressure multiplied by the unity matrix, K is the viscous stress tensor, F is the volume force vector, ρ is the fluid density, and u is the fluid velocity field.

The dielectrophoretic (DEP) force consists of the net polarization forces induced in a nonuniform electric field (NUEF) [52], where electrically neutral particles are asymmetrically polarized after exposure to the electric field. The resulting DEP force is 0 in uniform electric fields, while it is either positive or negative (pDEP or nDEP, respectively) in NUEF, as shown in Figure 1. This is achieved in the proposed design by the arrangement of electrodes. The DEP force equation is given by [63]:(5)$$F_{DEP}=2\pi \epsilon_{m}{R_{p}}^{3}Re\left[K\left(\omega \right)\right]\nabla \left|E_{0}^{2}\right|,$$
where $\epsilon_{m}$ is the fluid (medium) permittivity, $R_{p}$ is the affected particle radius, and $\nabla \left|E_{0}\right|$ is the electric field gradient. The electric field gradient is often the main control parameter of the DEP force, either through optimizing the geometry of the electrodes or by adjusting the applied voltage value. $K\left(\omega \right)$ is the Clausius–Mossotti factor (CMF), ω is the frequency of the applied signal, and $Re\left[K\left(\omega \right)\right]$ is the real part of the CMF. CMF denotes the polarization difference between the separated particles and the surrounding fluid [17,64]. It indicates whether a particle is repelled or attracted to the electrodes. The real part of CMF is given by the following expression [23]:(6)$$Re\left[K\left(\omega \right)\right]=\frac{\left(\epsilon_{p}-\epsilon_{m}\right)\left(\epsilon_{p}+2\epsilon_{m}\right)+\frac{\frac{\sigma_{p}-\sigma_{m}}{\omega }}{\frac{\sigma_{p}+2\sigma_{m}}{\omega }}}{{\left(\epsilon_{p}+2\epsilon_{m}\right)}^{2}+{\left(\frac{\sigma_{p}+2\sigma_{m}}{\omega }\right)}^{2}},$$
where $\epsilon_{p}$ is the particle permittivity, and $\sigma_{p}$ and $\sigma_{m}$ are the particles and medium electrical conductivities, respectively.

Using the COMSOL Multiphysics^®^ software package, the aforementioned physics are solved sequentially as follows:Solve for creeping flow;Solve for electric currents;Particle tracing based on the previous two physics.

It is important to note that this model neglects the two-way cell–cell interaction and is fully dependent on the fluid–cell interaction. This is a common limitation in modeling particle trajectories [27].

#### 2.2.2. Geometry

The proposed design is presented in Figure 2. It consists of two cascaded Y-junctions in series. The electrodes are located within the first stage, while the second stage is a passive one. The channel lengths, $C_{L1}$ and $C_{L2}$, are optimized to be as short as possible to reduce the footprint of the microchannel for packaging purposes. The channel width, $C_{W}$, is parametrically varied to study its effect on the separation of the cell species, while the depth, $C_{D}$, is fixed at 100 µm. The placement of the electrodes is not centered in the first stage of the microchannel to allow for flow focusing to bring the cells closer to the electrodes before attempting to separate. This is done to ensure that the DEP force is more effective, as it quickly loses its potency in distances further than ~30 µm [65,66,67]. The proposed LOC has a two-dimensional, planar design. Planar geometries have the added advantage of the ease of fabrication and being easy to implement in cases where multiple devices and/or stages need to be located on the same LOC, such as a CTC detection device [40]. A summary of the simulated ranges used for each of the geometric parameters is summarized in Table 2.

#### 2.2.3. Boundary Conditions

The no-slip Dirichlet boundary condition is applied on the walls of the LOC for the confined flow within the channel. For the buffer inlet, the flow rate is fixed to either 350 µm/s, 850 µm/s or 1350 µm/s. The blood inlet flow rate is set to 114 µm/s, 134 µm/s, or 154 µm/s. In contrast, the outlets have a static pressure boundary condition that is fixed at 0 Pa.

Fixed voltages, with values of $\pm V_{a}$, are applied on the electrodes in an alternating fashion. $V_{a}$ varies from ±2 up to ±4 V in 0.5 V increments. This is true for both the two– and four–electrode configurations of the LOC. The generated electric field induces the DEP force that affects the cell trajectories and eventually isolates them from one another.

The final step is to define the three cell types based on their properties introduced in Table 1. These cells are assigned to be released from the blood inlet simultaneously during the simulation window. The released particles are affected by two forces: internal drag forces and dielectrophoretic forces. The main limitation of this approach is the lack of particle–particle interaction modeling, as it only depends on the coupling between the fluid and the cells.

#### 2.2.4. Mesh Refinement

All obtained results in this study are produced on a computational grid shown in Figure 3 with inner triangular elements and quad elements on the boundaries. A grid independence test is carried out to evaluate three mesh element sizes denoted as “Coarse”, “Normal”, and “Fine” meshes based on the average element size.

The independence test for this model is based on the maximum errors, $\epsilon_{mesh}$, of the different cell velocities magnitude at the target outlet of each of the respective cell types. $\epsilon_{mesh}$ is the relative error between successive mesh refinement iterations. An identical microchannel setup was used for each of the considered meshes and $\epsilon_{mesh}$ was then compared to a criterion $\epsilon_{criterion}=1\%$. The calculated errors are summarized in Table 3. Our results indicate that a fine mesh with a maximum error of 0.15% is adequate for this model. 

### 2.3. Numerical Model Validation

The model is validated by comparing it to the experimental results of Piacentini et al.’s [16] setup. The comparison is based on RBC and blood platelet (PLT) cell trajectories, as shown in Figure 4. The developed model uses identical parameters to the experimental setup: an applied voltage of ±5 V, a fixed cell inlet velocity of *v_in,cells_* = 134 µm/s, and a buffer inlet velocity of *v_in,buffer_* = 853 µm/s. The experimental cell locations are superimposed onto the simulated cell trajectories, showing an agreement between the simulation and the experimental results. Therefore, the model is validated.

### 2.4. Simulated Design Parameters

Parametric studies based on the dimensions of the microchannel are simulated, and the cell separation efficiencies and purities are evaluated for every configuration. Starting from a base velocity of *v_in,buffer_* = 850 µm/s taken from the literature [16], the inlet velocity is varied parametrically in 100 µm/s increments above and below the base velocity. The highest and lowest velocities that achieved full cell separation are *v_in,buffer_* = 1350, and *v_in,buffer_* = 350 µm/s, respectively. Hence, they are selected as the upper and lower limits for the tested configurations. The cell inlet velocity is set to *v_in,cells_* = 134 µm/s for the two-electrode design, while the four-electrode designs are varied between *v_in,cells_* = 114, 134, and 154 µm/s. The different combinations of the simulated parameters are summarized in Table 4.

## 3. Results

This section defines the main metrics used to evaluate the performance of the LOC, such as the separation efficiency for each cell type, the purity of each outlet, and the throughput of the device. The impact of changing each of the buffer and cell inlet velocities, channel width, electrode configuration and applied voltage is also investigated. Drag forces, fluid velocity profile, and pressure contours are also presented.

### 3.1. Separation Efficiency

Separation efficiency is the most used metric for evaluating separator LOCs. It is defined as the number of isolated target particles (cells), to the number of input target particles (cells) as a percentage [3]. Separation efficiency is evaluated by the following expression:(7)$$\eta_{Separation}=\frac{{No.\;of\;targeted\;cells}_{isoltated}}{{No.\;of\;targeted\;cells}_{injected}}\times 100\%.$$

The effectiveness of different configurations of the proposed design is evaluated based on this metric per-cell type, with the ones with 100% efficiency for all cells of the simulated configurations summarized in Table 5. Separation efficiencies for each cell type are individually listed in the Appendix A. An example of a two-electrode design with the following velocities, namely, *v_in,buffer_* = 850 µm/s, and *v_in,cells_* = 134 µm/s, is shown in Figure 5, and as an animation in Video S2.

### 3.2. Purity

Purity is the metric used to assess the performance of the LOC based on the output of each outlet. It is defined as the ratio of the targeted target particles (cells) at the specified outlet to the number of all particles (cells) at the same outlet as a percentage. It is calculated by the following expression for each outlet [3]:(8)$$Purity={\left(\frac{No.\;\;of\;targeted\;cells}{No.\;\;of\;targeted\;cells+No.\;\;of\;unwanted\;cells}\right)}_{desired\;outlet}\times 100\%$$

The aim is to have as high of purity as possible, as lower purity values indicate the existence of contaminants in the output. Configurations with 100% purity for all outlets are shown in Table 5, while the purity values of each outlet for all tested configurations are found in the Appendix A.

### 3.3. Dielectrophoretic Force

The dielectrophoretic force is calculated using COMSOL Multiphysics^®^ for the same type of cells in the microchannel during an 8-s simulation interval using Equation (5). The DEP force changes over the simulation time are shown in Figure 6. WBCs are the most influenced cell species by the DEP force, leading to their removal from the outlet furthest from the electrodes. In a similar fashion, PLTs (affected by the weakest DEP) exit from the outlet closest to the electrodes. The DEP force effect lasts for approximately 5.5 s (1 to 6.5 s), during which the particles are within the first stage. The DEP force drops for all cells as soon as they exit the active stage. The drop before 1 s occurs while the cells enter the LOC from the cell inlet.

### 3.4. Fluid Velocity and Pressure

Figure 7 visualizes the velocity profile for the four-electrode configuration. The buffer inlet velocity of the lower inlet is set to either high (1350 µm/s) or low (850 µm/s) velocity in Figure 7a,b, respectively, while the cell inlet velocity of 134 µm/s is used in both instances. As *v_in,buffer_ > v_in,cells,_* the buffer focuses the cell-carrying blood closer to the electrodes, ensuring that the CTCs, WBCs, and PLTs fall within the NUEF region. This behavior is also apparent after the first junction, where the lower outlet has higher velocity magnitudes on average compared to the upper one. It is at this stage that the velocity profile becomes uniform across the channel width. The velocity magnitude decreases until it reaches its minimum value at the end of the second stage.

Lower pressures in the microchannel are generally preferable to ensure that blood cells retain their shape, as higher pressures can cause cell deformations [68]. Investigating pressure distribution shows that configurations with higher buffer inlet velocity have pressure value that is higher than configurations with lower buffer inlet velocity, as can be seen in Figure 8. In the case of two identical setups, except for the buffer inlet velocity, the higher inlet velocity configuration has a higher pressure at the inlet compared to the lower inlet velocity configuration. The pressure falls to 0 Pa at the outlets, as enforced by the boundary conditions.

## 4. Discussion

This section discusses the influence of DEP force on each cell type. Based on the obtained results, the effects of each of the buffer flow rates, the number of electrodes, the applied voltage, and the channel geometry on the separation efficiency are then discussed. The definition of the throughput is also introduced for microfluidic systems. A summary of the separation efficiency for each species, as well as the purity of the outlets for all tested configurations, is provided in the Appendix A. Finally, the optimal design for complete cell separation is proposed.

### 4.1. Dielectrophoretic Force

Dielectrophoretic (DEP) force is the main mechanism responsible for cell separation in the proposed design as it affects the trajectories of the cells. We can observe from Figure 6 that, on the one hand, WBCs are affected with the largest DEP force, and hence, that these types are the first to exit the microchannel from the lower outlet (furthest from the electrodes). On the other hand, PLTs have the smallest DEP force, so they tend to be closer to the electrodes exiting from the upper outlet in the microchannel. Similarly, CTCs exit the microchannel from the middle outlet as they are affected by the DEP force that falls between the other two. This behavior is confirmed by the trajectories of the cells in Figure 9a,b. The gaps between the path of the cells and the walls of the channel outlets are denoted as α, β, and γ for WBCs, CTCs, and PLTs, respectively. The two Appendix A, demonstrate the particle trajectories for four- and two-electrode configurations, respectively.

### 4.2. Throughput

Throughput is another metric that measures how quickly the separation process is carried out. It is typically measured as the volumetric flow rate at the desired outlet, but it is useful in some circumstances to multiply the flow rate value by cell density per volume to account for dilution during sample preparation [69]. Alternatively, the number of sorted cells can be reported. The throughput value is often not critical in microfluidic applications as it can be increased by the use of parallel microchannels [70,71].

### 4.3. Impact of Cell and Buffer Inlet Velocities on the Separation Efficiency

Increasing the buffer velocity up to a certain point generally requires higher voltage values to achieve comparable separation efficiencies. An additional increase in the buffer inlet velocity will compromise the separation of CTCs from the rest of the blood components in most cases, as the main flow is focused closer to the electrodes.

The effect of decreasing the buffer inlet velocity from *v_in,buffer_* = 1350 µm/s to *v_in,buffer_* = 850 µm/s is shown in Figure 9a,b, respectively. The impact of changing *v_in,buffer_* is apparent in the trajectories of the cells, where higher *v_in,buffer_* values cause the cell to approach the upper wall (electrode-side) more closely, as demonstrated by the distances between the cells and the walls of the outlets Figure 9. This leads to WBCs and CTCs being closer to the upper wall of the outlet. An additional increase in *v_in,buffer_* forces a portion of the WBCs to skip the outlet altogether, compromising the separation efficiency. Moreover, using high buffer inlet velocity necessitates increasing the electrode voltage range, risking cell damage. Similarly, lower buffer inlet velocity *v_in,buffer_* = 350 µm/s did not allow the targeted cells to get close enough to the NUEF, and no separation could be achieved in this instance.

As for cell inlet, changing *v_in,cell_* did not have a significant effect on the separation performance in most circumstances. The exception to this occurred at relatively high *v_in,cell_* = 154 µm/s, and lower *v_in,buffer_* as the buffer fails to focus the blood flow effectively in these circumstances. Using lower cell inlet velocity can also reduce the overall throughput of the microchannel, but this can be easily addressed by using parallel microchannels to increase the throughput.

### 4.4. Impact of Electrodes Configuration and Applied Voltage on the Separation Efficiency

The electric field distributions of four- and two-electrode configurations are shown in Figure 10. Applied voltage polarity alternates between adjacent electrodes, as seen in Figure 10. The most important characteristic is that a non-uniform electric field is generated, giving rise to the dielectrophoretic force that is responsible for the separation of the particles. The applied electrode voltage ranges from ±2 to ±4 V, giving a peak-to-peak range of 4–8 V. Higher voltage values push the cell trajectories away from the electrodes, as apparent in simulation results in the Appendix A. Considering the simulated combinations, the wider channels require higher applied voltage values to achieve similar separation performances.

Using fewer electrodes would require that a higher voltage value be used to obtain the same efficiency. In some cases, increasing the voltage is not sufficient without using unreasonably high voltage values. Additionally, using two-electrode configurations has noticeably lower separation efficiencies compared to four-electrode configurations. Both shortcomings for the two-electrode configurations are visible in Table 5 and Appendix A compared to the alternative four-electrode designs. At this point, using more electrodes (four electrodes) at lower voltages proved to be the better approach, especially when considering the possibility of cell damage at higher voltages.

### 4.5. Impact of Changing Channel Width on Electrode Voltage

One of the design objectives is to minimize the applied voltage value to protect the target cells from potential membrane damage. For this reason, using narrow channels is more advantageous compared to wider channels, as the width of the microchannel greatly affects the required voltage values. Generally, wider channels are found to require higher applied voltage values to obtain a comparable separation efficiency to channels with narrower widths. For four-electrode configurations, it is especially apparent at higher buffer inlet velocities, namely *v_in,buffer_* = 850 µm/s, as demonstrated in Table 5.

### 4.6. Optimum Design

The proposed design is selected from the simulated design configurations summarized in Table 4, prioritizing the use of low voltage potentials for minimal cell membrane damage. First, four-electrode configurations are preferred due to their lower voltage requirement. The lowest voltage that achieved complete separation is ±2 V. Narrower channels are preferred, as previously mentioned, due to the higher voltage requirement of wider channels. Finally, a buffer inlet velocity of at *v_in,buffer_* = 850 µm/s is selected. The chosen parameters are given in Table 6, and separation performance is demonstrated in the cell trajectories in Figure 11.

## 5. Conclusions

A two-stage microchannel LOC capable of separating WBCs, PLTs, and CTCs is designed and simulated. The proposed design uses dielectrophoretic force (DEP) in conjunction with flow focusing to isolate each cell type based on its size and electrical properties. This approach can efficiently separate CTCs from other blood cells. The LOC can be cascaded with other devices, such as cell counters. Moreover, the device uses low voltage values, minimizing potential cell damage. The geometry of the LOC is parametrically optimized with the aim of achieving a high separation efficiency for all three target cells. The impact of adjusting the number of electrodes and voltage intensity, channel geometric parameters, and buffer inlet velocities on the separation efficiency and cell trajectories can be summarized as follows:Increasing the buffer inlet velocity, *v_in,buffer_*, can compromise the trajectories of the target cells by forcing the cells to move closer to the electrode and, hence, reducing the overall separation efficiency;The use of microchannels with wider widths requires an increase in the applied voltage value to achieve comparable levels of efficiency;Using four electrodes allows the usage of lower voltage values compared to using just two electrodes;Higher voltage values induce a stronger DEP force that forces the cells to move further from the electrodes.

As such, the proposed design aims to achieve full separation while using a relatively low voltage of ±2 V to minimize cell membrane damage.

This study can potentially be expanded by investigating the separation of other blood cell types. Additionally, the functionality of the LOC can be improved by adding cell counters. More realistic modeling of non-spherical cells can also be considered. Finally, modeling the particle–particle interaction can improve the reliability of the simulation instead of fully depending on just the fluid–particle interaction.

## Figures and Tables

**Figure 1 micromachines-14-01769-f001:**
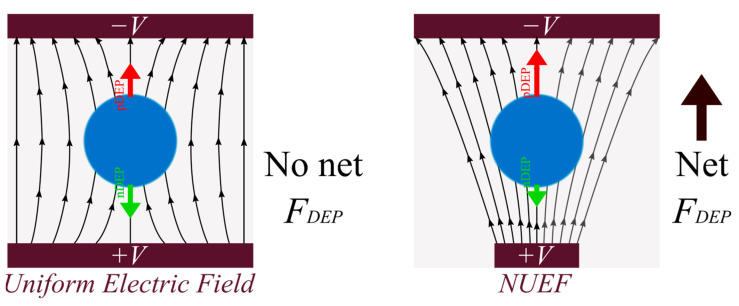
Electrophoresis effect on neutral particles in uniform and non-uniform electric fields.

**Figure 2 micromachines-14-01769-f002:**
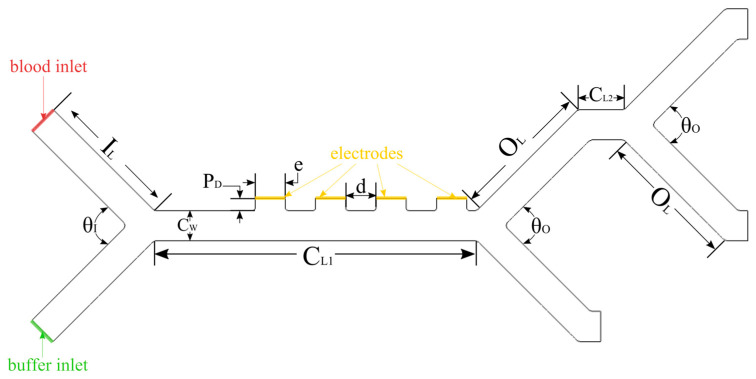
Separator microchannel structure and terminology for the design dimensions. The overall size is about 919 µm × 440 µm × 100 µm, and all inlets and outlets have the same length.

**Figure 3 micromachines-14-01769-f003:**
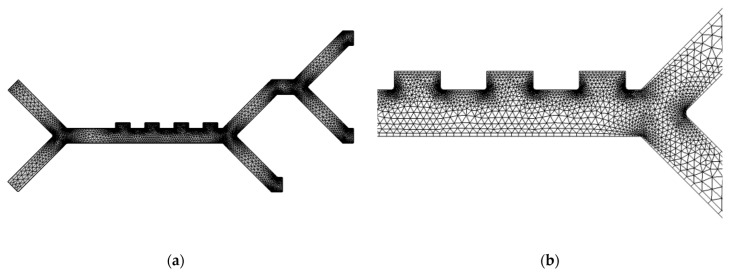
Mesh of fine element size showing: (**a**) Overview of the entire geometry; (**b**) close-up view of the Y-channel and electrodes, showing the distribution of the triangular and quad mesh elements.

**Figure 4 micromachines-14-01769-f004:**
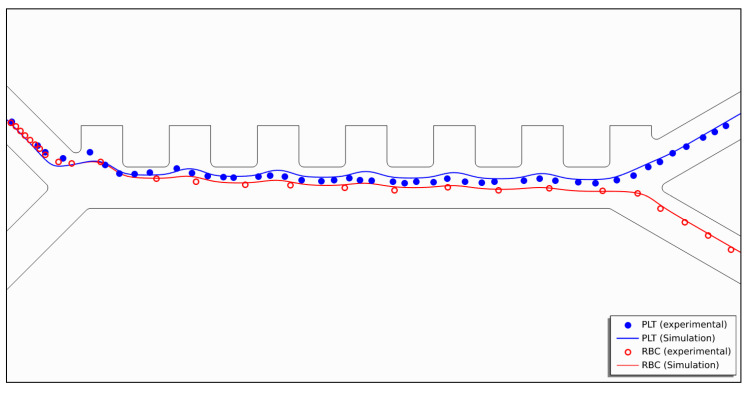
Validation of the numerical model showing simulated cell trajectories for RBCs (red) and PLTs (blue), with the experimental results of Piacentini et al.’s superimposed on top of it.

**Figure 5 micromachines-14-01769-f005:**
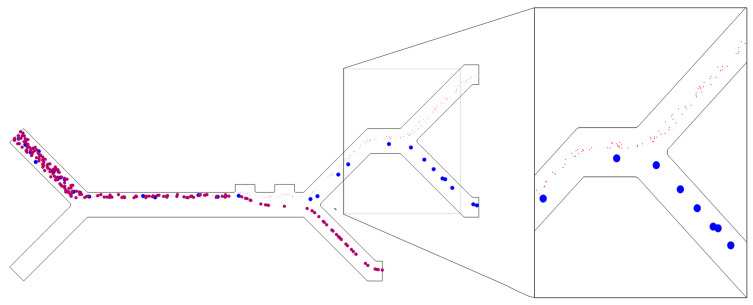
Cell trajectories for CTCs (blue), PLTs (red), and WBCs (purple). This configuration shows 100% separation efficiencies for the three cell species using two electrodes at ±4.0 V, with a close-up view showing PLTs and CTCs at the upper Y-split. The inlet velocities are *v_in,buffer_* = 850 µm/s, *v_in,cells_* = 134 µm/s, and *C_w_* = 50 µm.

**Figure 6 micromachines-14-01769-f006:**
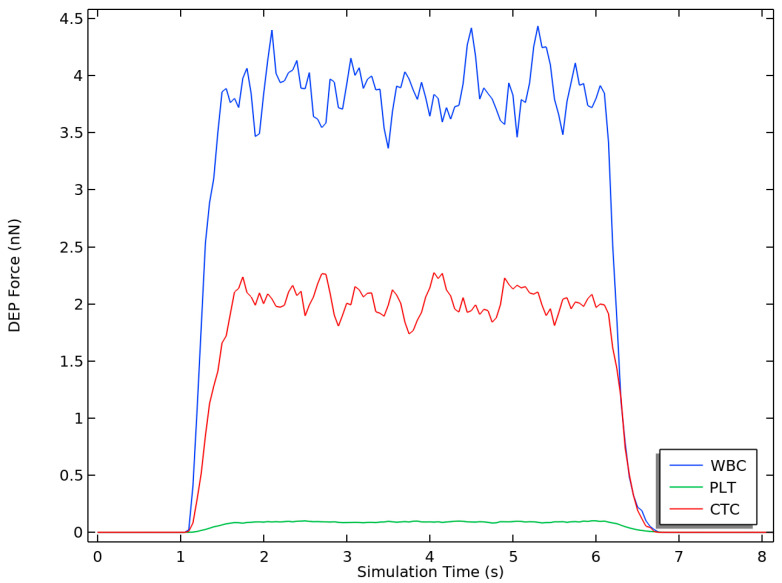
Dielectrophoretic force for each cell species evaluated over the simulation time. The following configuration was used: four electrodes, ±3.5 V, *C_w_* = 50 µm, and inlet velocities of *v_in,buffer_* = 1350 µm/s and *v_in,cells_* = 134 µm/s.

**Figure 7 micromachines-14-01769-f007:**
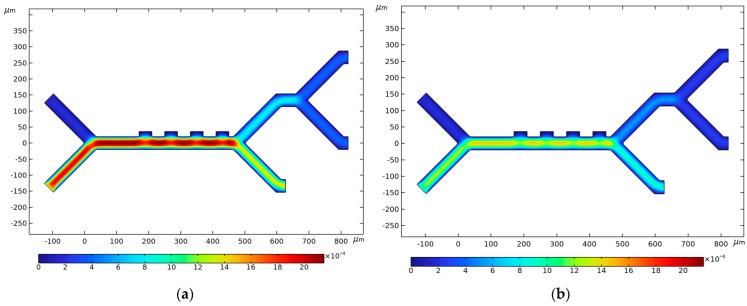
Fluid velocity magnitude in µm/s for: (**a**) *v_in,buffer_* = 1350 µm/s, and *v_in,cells_* = 134 µm/s; (**b**) *v_in,buffer_* = 850 µm/s, and *v_in,cells_* = 134 µm/s. Channel width in both instances is *C_w_* = 40 µm.

**Figure 8 micromachines-14-01769-f008:**
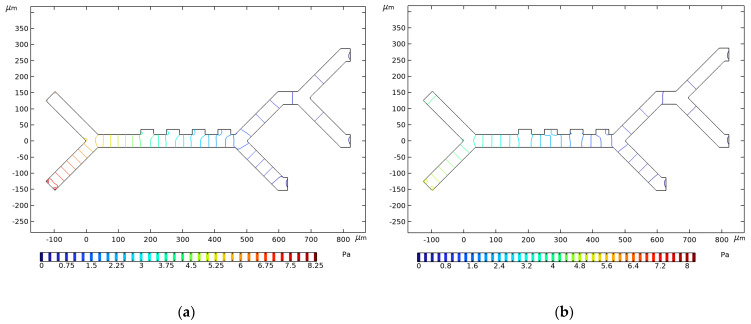
Pressure isolines in Pa for (**a**) *v_in,buffer_* = 1350 µm/s, and *v_in,cells_* = 134 µm/s; (**b**) *v_in,buffer_* = 850 µm/s, and *v_in,cells_* = 134 µm/s. Channel width in both instances is *C_w_* = 40 µm.

**Figure 9 micromachines-14-01769-f009:**
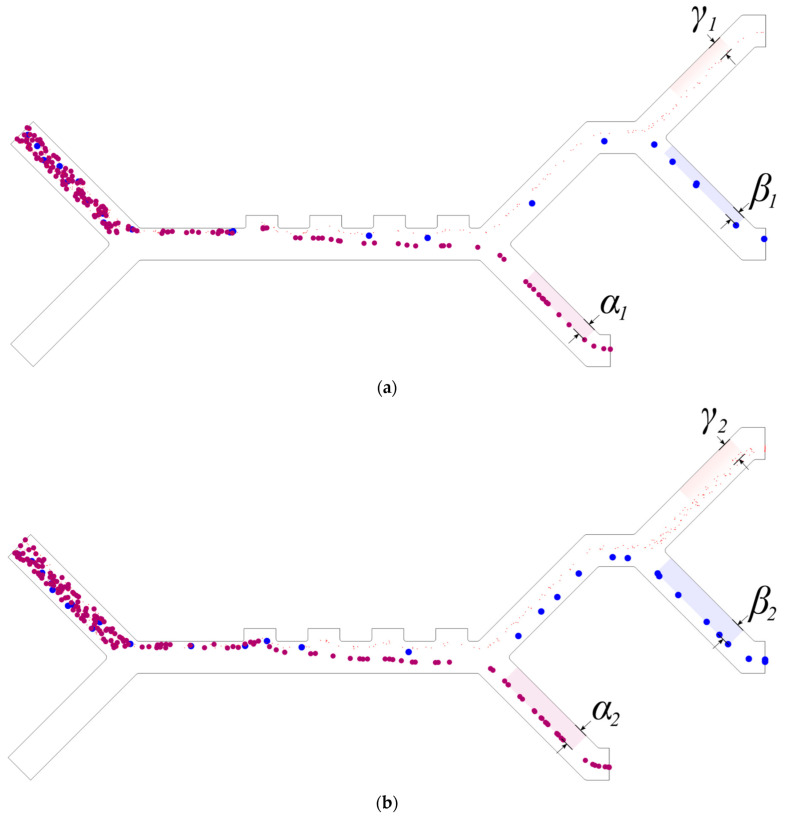
Cell trajectories for CTCs (blue, *β*), PLTs (red, γ), and WBCs (purple, *α*) for two 4-electrode configurations at ±2.5 V, *C_W_* is 40 µm, and *v_in,cells_* is 134 µm/s. The gaps between the cells and the outlet wall (shaded areas) increase with lower buffer velocities, so *α_1_ < α_2_*, *β_1_ < β_2_*, and *γ_1_ < γ_2_*: (**a**) *v_in,buffer_* = 1350 µm/s; (**b**) *v_in,buffer_* = 850 µm/s.

**Figure 10 micromachines-14-01769-f010:**
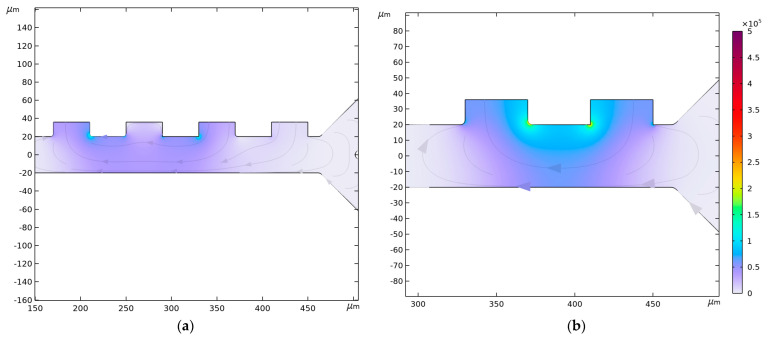
Electric field distribution in V/m around the electrodes showing the non-uniform electric field (NUEF) in (**a**) four-electrode configuration at ±4 V; (**b**) two-electrode configuration at ±4 V. Channel width in both instances is *C_w_* = 40 µm.

**Figure 11 micromachines-14-01769-f011:**
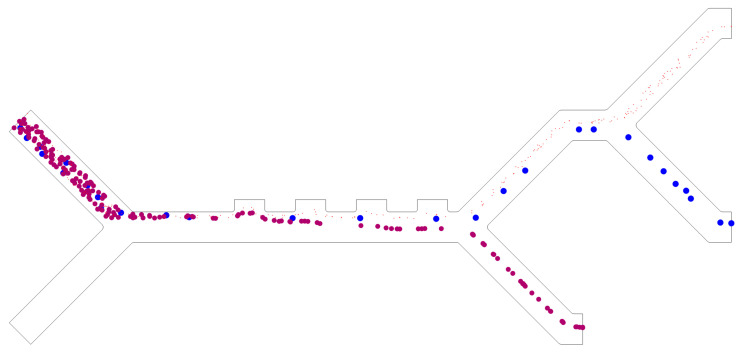
Cell trajectories for CTCs (blue), PLTs (red), and WBCs (purple) for the proposed design. 4 electrodes at ±2.0 V, *C_W_* = 40 µm, *v_in,buffer_* = 850 µm/s, and *v_in,cells_* = 134 µm/s.

**Table 1 micromachines-14-01769-t001:** Physical and electrical properties of target cells.

Parameter	CTC	WBC	PLTs
Cell diameter	15 µm	12 µm	1.8 µm
Cell membrane thickness	7 nm	7 nm	8 nm
Cell conductivity	1 S/m	0.18 S/m	0.25 S/m
Membrane conductivity	9 × 10^−7^ S/m	9 × 10^−6^ S/m	1 × 10^−6^ S/m
Cell relative permittivity	50	80	50
Membrane relative permittivity	12.5	10	6

**Table 2 micromachines-14-01769-t002:** Major dimension for simulated design configurations.

Dimension	Value	Description
*I_L_*	190 µm	Inlet length
*C_L_* _1_	500 µm	Main channel length
*C_L_* _2_	60 µm	Secondary channel length
*C_W_*	40, 50, 60 µm	Channel width
*C_D_*	100 µm	Channel depth
*O_L_*	190 µm	Outlet length
*θ_I_*	90°	Inlet channels angle
*θ_O_*	90°	Outlet channels angle
*P_D_*	16 µm	Electrode protrusion depth
*e*	40 µm	Electrode width
*d*	40 µm	Distance between electrodes

**Table 3 micromachines-14-01769-t003:** Mesh refinement comparison of the three mesh configurations, coarse, normal, and fine, based on the velocities of the different cell types at their respective outlets.

Cell Type	Coarse and Normal Meshes	Normal and Fine Meshes
PLT	50.59%	0.06%
CTC	0.26%	0.08%
WBC	395.24%	0.15%
$\epsilon_{mesh}\leq \epsilon_{criterion}$	No	Yes
$\epsilon_{mesh}$ **value**	395.24%	0.15%

**Table 4 micromachines-14-01769-t004:** Summary of the different combinations used in simulating the LOC designs.

*Four-Electrode Variant* *v_in,buffer_* = 350 µm/s, 850 µm/s, 1350 µm/s *v_in,cells_* = 114 µm/s, 134 µm/s, 154 µm/s		*Two-Electrode Variant* *v_in,buffer_* = 350 µm/s, 850 µm/s, 1350 µm/s *v_in,cells_* = 134 µm/s	
*Main Channel Width*	*Applied Electrode Voltage*	*Main Channel Width*	*Applied Electrode Voltage*
40 µm	2.0 V	40 µm	2.0 V
40 µm	2.5 V	40 µm	2.5 V
40 µm	3.0 V	40 µm	3.0 V
40 µm	3.5 V	40 µm	3.5 V
40 µm	4.0 V	40 µm	4.0 V
50 µm	2.0 V	50 µm	2.0 V
50 µm	2.5 V	50 µm	2.5 V
50 µm	3.0 V	50 µm	3.0 V
50 µm	3.5 V	50 µm	3.5 V
50 µm	4.0 V	50 µm	4.0 V
60 µm	2.0 V	60 µm	2.0 V
60 µm	2.5 V	60 µm	2.5 V
60 µm	3.0 V	60 µm	3.0 V
60 µm	3.5 V	60 µm	3.5 V
60 µm	4.0 V	60 µm	4.0 V

**Table 5 micromachines-14-01769-t005:** Summary of the design configurations with complete separation (achieving both 100% purity for all outlets and 100% separation efficiency for CTCs, PLTs, and WBCs).

*v_in,buffer_*	850 µm/s		1350 µm/s		*Separation* *Efficiency*	*Purity (All Outlets)*
*Number of Electrodes*	*Channel Width*	*Electrode Voltage*	*Channel Width*	*Electrode Voltage*		
4	40 µm	2.0, 2.5 V	40 µm	2.5, 3.0, 3,5 V	100.00%	100.00%
4	50 µm	2.5, 3.0 V	50 µm	2.5 *, 3.0, 3.5, 4.0 V		
4	60 µm	2.5 **, 3.0, 3.5, 4.0 V	60 µm	3.5, 4.0 V		
2	40 µm	3.5, 4.0 V	40 µm	4.0 V		
2	50 µm	3.5, 4.0 V	50 µm	N/A		

* Only for *v_in,cell_* = 114 µm/s, and *v_in,cell_* = 134 µm/s. ** Only for *v_in,cell_* = 114 µm/s.

**Table 6 micromachines-14-01769-t006:** The main parameters of the proposed design.

Parameter	Value	Description
*N*	Four electrodes	Number of electrodes
*V_a_*	±2.0 V	Applied voltage
*C_W_*	40 µm	Channel width
*v_in,buffer_*	850 µm/s	Buffer inlet velocity
*v_in,cells_*	134 µm/s	Cell inlet velocity

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
