# Peer review of "Numerical Simulation of a Lab-on-Chip for Dielectrophoretic Separation of Circulating Tumor Cells"

_micromachines, 2023, doi:10.3390/mi14091769_

Round 1

Reviewer 1 Report

The authors present a simulation-based study of a purportedly novel electrode platform for three-stage dielectrophoretic separation.  There are a number of issues with the work which would need to be addressed before publication can be considered.

 The primary concern with the paper is its (repeatedly stated) aim of separating CTCs from blood.  The authors correctly state that there is a need for separation of CTCs and that DEP could do this – however, this application was first demonstrated a decade ago by Gupta et al. in Biomicrofluidics.  Their platform – the ApoStream – has been commercially available for years and can be seen here: https://www.precisionformedicine.com/specialty-lab-services/tissue-liquid-biopsy/apostream-ctc-and-rare-cell-isolation-for-liquid-biopsy/.   Furthermore, the electrode geometry used is not hugely dissimilar to that presented here – consisting of two inlet s and two outlets, with electrodes on one wall (or in the ApoStream, the floor) of the device to push cells out of one outlet flow and into another. 

 The simulations are based on an electrode design from 2011, which is fine, but that paper described a platelet separator rather than a separator for CTCs.  If a separator is to be designed for that application, the authors need to be aware of the magnitude of the problem.  As the authors correctly state, a liquid biopsy typically contains 1-20 CTCs.  The same sample contains about 2 million white blood cells and 2 billion red blood cells.  Eliminating the red cells by centrifugation still leaves a separation of 1 cell in 2 million – or more likely 5 cells in 10 million to reduce the chance of missing/losing that one vital cell.  Sorting the RBCs as proposed here is a non-starter I’m afraid – you can’t do DEP separation at cell concentrations analogous to blood, so you’re looking at separating perhaps 100ml of solution to get to your 1 ml of blood.  Given the fastest DEP sorters go at 1 ml/min, the proposed system looks unlikely to offer any benefit.

 Realistically that means sorting upwards of 10 ml of WBC solution with very high accuracy, so two things matter here: throughput (ml/min) and sensitivity.  Neither are addressed here.     The flow rate is described in m/s – which means little in pressure-driven flow where different flow rates are experienced across the channel.  It is far better to discuss flow rates in ul/sec, since this allows the reader to understand the device in practical terms – for example, how long it will take to sort 1 ml of cell solution at concentration 1 million/ml.    For accuracy, what is required is an analysis of how different cell types will respond to the field.  There will be variation in where the cells will come out of the inlet pipe – some will be nearer, others further – and there is a natural degree of variation in cell parameters (radius etc).  It would be useful to have done some kind of analysis of the likelihood of WBCs ending up in the CTC outlet to give an indication of how effective the enrichment would be. 

 I would suggest that for this paper to be publishable, it need to be refocussed on either making it more realistic (and that is possible, as ApoStream demonstrated) in terms of the accuracy of the output, throughput and so forth (in particular, removing the RBC stage which will not work given the application); or conversely to lose the focus on CTCs and make the paper a more general one on device optimisation.  Note incidentally that the authors need to be better acquainted with the literature; there are a number of electrodes which do similar sorts, such as Wang et al Electrophoresis2009,30, 782–791782, including 3-way sorts such as this: https://pubs.aip.org/aip/bmf/article/13/6/064111/238877.  The simulated performance of your device should at least be benchmarked against these, with some kind of indication as to why your work represents an advance over their work.

 Minor point on page 2.  More than half the page is given over to describing device fabrication, which is redundant in a simulation paper where you did no fabrication.

Author Response

Attached response

Reviewer 2 Report

This manuscript describes a COMSOL simulation work on using DEP in a LOC device to separate circulating tumor cells. The results are sound and well-presented. The paper is quite detailed in its simulation effort. I believe it can be a valuable addition to Micromachines and can be a useful study case of using COMSOL in LOC applications. I recommend publication of this manuscript. 

Author Response

Attached response

Reviewer 3 Report

This work proposed a two-stage Y-junction microfluidic design as a LOC device for dielectrophoretic separation of CTC, WBC and RBC from whole blood samples. COMSOL was employed for parameter optimization to achieve a high separation efficiency with low applied voltage. The work provides insights and guidance for rational design of microfluidic devices that apply dielectrophoretic separation for particle separation. The manuscript can be considered to be published after making the following changes:

1. There have been many reports both on experiments and simulations for separating CTC from whole blood including microfluidic systems applying electrophoresis. Please provide a more thorough review on those closely related works in the introduction section. The concept and application of dielectrophoresis field-flow-fractionation (DEP-FFF) should also be discussed here when proposing the microchannel design. Some of the papers that should be cited:

https://doi.org/10.1016/j.jsamd.2021.03.005

https://doi.org/10.1039/C0LC00345J

https://doi.org/10.1016/j.ebiom.2022.104237

2. Section 2.2.4: please specify which mesh was selected for simulations and why is 2% selected as the criterion (instead of 1% or 5% etc.). This criteria selection seems to be arbitrary. Please provide more context.

3. Where would platelets and other cells/particles in the blood samples end up being after the separation process? Please provide more context on the sample pre-process (e.g. centrifuge) and/or post-process if applicable in this proposed design. 

4. How does inlet cell velocity influence the separation results? 134um/s was used in the experiments (citation [17]) for platelets separation so it might not be the optimal inlet cell velocity for the CTC separation purpose described in this work. This parameter should be discussed in more details. The same question applies to the buffer velocity of 835 um/s. Does deceasing the buffer velocity provide any benefits in the setup presented in this work?

5. Figure 3: The four dead-end channels that are next to the electrodes are not evenly distributed along CL1, i.e. the first electrode is further to the left end of CL1 comparing to the last electrodes to the right end of CL1. What is the rational of this design? How does the location of electrodes along CL1 impact the separation?

6. Table 5: providing additional separation efficiencies for other conditions described in Table 4 maybe helpful for trend observation and guide the rational design of similar devices.

7. Figure 8 and 9: figure 8 shows it takes 4 seconds for all the cells to enter the channel, while figure 9 indicates that the DEP drops to zero before 3.5 second. Is this because only the CL1 is an active separation stage? How does time translate to how long that the particles travel in the microfluidic channel? Please provide more information on the relation between time and distance.

8. Supplemental materials were not mentioned in the manuscript. Please add in referral and description at where those results are discussed.

Author Response

Attached response

Round 2

Reviewer 1 Report

The paper is much improved, and could be acceptable in its current form, with one caveat; the removal of the RBC outlet is sensible and I applaud that, but the design is now left with a dividing channel with only one side being used; it would make more sense if the design were simply truncated after the first split, with a white box removing the second one.  If this were done the figures would make more sense and the paper would be better received.  Other than this I think the paper is fine.
